# Myalgic Encephalomyelitis/Chronic Fatigue Syndrome (ME/CFS): Major Impact on Lives of Both Patients and Family Members

**DOI:** 10.3390/medicina57010043

**Published:** 2021-01-07

**Authors:** Esme Brittain, Nina Muirhead, Andrew Y. Finlay, Jui Vyas

**Affiliations:** 1School of Medicine, Cardiff University, Cardiff CF14 4XN, UK; BrittainEL@cardiff.ac.uk (E.B.); FinlayAY@cardiff.ac.uk (A.Y.F.); 2Buckinghamshire Healthcare NHS Trust, Amersham Hospital, Amersham HP7 0JD, UK; nina.muirhead@btinternet.com

**Keywords:** ME/CFS, QoL, family impact, FROM-16, WHOQOL-BREF

## Abstract

*Background and objectives:* To explore the impacts that Myalgic Encephalomyelitis/Chronic Fatigue Syndrome (ME/CFS) has on the patient and their family members using the WHOQOL-BREF (Abbreviated World Health Organisation Quality of Life questionnaire) and FROM-16 (Family Reported Outcome Measure-16) quality of life assessments. *Materials and Methods:* A quantitative research study using postal questionnaires was conducted. A total of 39 adult volunteers expressed an interest in participating in the study: 24 returned appropriately completed questionnaires. Patients with ME/CFS completed the WHOQOL-BREF and up to four of their family members completed the FROM-16 questionnaire. *Results:* ME/CFS negatively affects the quality of life of the patient (median scores WHOQOL-BREF: Physical health = 19, Psychological = 44, Social relationships = 37.5, Environment = 56, *n* = 24) and their family members’ quality of life (FROM-16: Emotional = 9.5, Personal and social = 11.5, Overall = 20.5, *n* = 42). There was a significant correlation between the patient’s reported quality of life scores and their family members’ mean FROM-16 total scores. *Conclusions:* This study identifies the major impact that having an adult family member with ME/CFS has on the lives of partners and of other family members. Quality of life of ME/CFS patients was reduced most by physical health compared to the other domains. Quality of life of family members was particularly impacted by worry, family activities, frustration and sadness. This highlights the importance of measuring the impact on the lives of family members using tools such as the FROM-16 in the ME/CFS clinical encounter and ensuring appropriate support is widely available to family members.

## 1. Introduction

Myalgic Encephalomyelitis/Chronic Fatigue Syndrome (ME/CFS) has profound impacts on the lives of those affected, but little is known about the impact experienced by partners and other family members. This hidden family burden is often ignored or unrecognised by health care workers.

ME/CFS is a complex, multisystem disease involving neurological, immunological, autonomic, and energy metabolism impairments [1]. Symptoms include post-exertional malaise (PEM), orthostatic intolerance, cognitive difficulties and unremitting fatigue [2]. ME/CFS is diagnosed based on clinical criteria, due to the absence of a known biomarker. The aetiology is unclear and there is no definitive pharmacological treatment; current treatment options target symptoms, rather than an underlying cause [2]. This lack of knowledge and evidence concerning diagnosis, aetiology and therapy add an additional burden to the practical issues experienced by those living with someone with ME/CFS.

Health-related quality of life (QoL) in patients with ME/CFS is significantly lower than in healthy controls and patients with other chronic illnesses [3]. Some studies have explored ME/CFS in paediatric patients and the impact on their mothers [4] and siblings [5]. However, there are no studies to date which have explored the impact of ME/CFS on both adult patients’ QoL and their family members’ QoL using validated questionnaires.

It is important to identify the extent to which the QoL of partners and family members of adults with ME/CFS is affected. Such knowledge may appropriately influence management decisions and also highlight areas of support that are required for both the patient and the family members. Measurement of QoL may thereby have the potential to enhance the quality of care of patients and their families.

The aim of the study was to measure the impact of ME/CFS on the patient and family using the WHOQOL-BREF (Abbreviated World Health Organisation Quality of Life questionnaire) and FROM-16 (Family Reported Outcome Measure-16) questionnaires. In families where patients reported a poorer QoL, we hypothesised that there would be a significant impact on the family member’s QoL.

## 2. Materials and Methods

Ethical approval was granted by the Cardiff University School of Medicine Research Ethics Committee on 1 March 2019 (reference number 19/29).

Information regarding the study was posted on the Welsh Association of ME & CFS Support charity website and social media pages, through which patients made contact. Patient volunteers who expressed an interest in participating were sent packs in the post containing: research information leaflets, consent forms, one WHOQOL-BREF questionnaire for the patient and four FROM-16 questionnaires for up to four of their family members. Participants were excluded if the patient did not have a formal diagnosis of ME/CFS, if questionnaires were incomplete, or if the patient or family member were under the age of 18 years. All included patients and family members consented to participating in the study.

Participant information and questionnaire responses were recorded on separate password-protected Excel spreadsheets on a Cardiff University computer. WHOQOL-BREF questionnaires included a two-digit code number and letter A (e.g., 01 A). Each FROM-16 questionnaire included a three-digit code and letter B (e.g., 01.1 B). Each family received a different number. This coding ensured responses remained anonymous but still grouped when interpreting the data.

The WHOQOL-BREF is a 26-item questionnaire, which measures the impact of an illness on a patient’s QoL [6] and is suitable for measuring the impact of ME/CFS [7]. Patients are assessed across four domains: Physical Health, Psychological, Social Relationships and Environment. A lower score in the WHOQOL-BREF indicates a poorer QoL. This questionnaire includes two additional questions regarding the patient’s overall perception of their QoL and their health satisfaction. Each question has a five-point Likert interval scale where patients rate their response from ‘Not at all’ (1 point) to ‘An extreme amount’ (5 points). Patients were also asked to self-time how long it took to answer the WHOQOL-BREF.

The FROM-16 is a 16-item questionnaire designed to assess the impact of a disease on the patient’s partner and family members [8]. The maximum score is out of 32, with a higher score indicating a greater impact on the family member’s QoL. The questionnaire consists of two domains: Emotional (Part 1) and Personal and Social Life (Part 2). Each answer is graded on a three-point scale consisting of ‘Not at all’, ‘A little’ and ‘A lot’. Statistical averages (mean, median, range, standard deviation) were used to assess the data collected by the questionnaires. Spearman’s rank correlation coefficient was used to measure the strength of the correlation between different aspects of the WHOQOL-BREF and the mean FROM-16 total scores.

## 3. Results

Of 39 questionnaires posted in response to expression of interest, 29 were returned, giving a response rate of 74%. Five were excluded from analysis due to incomplete questionnaires and a self-reported diagnosis, resulting in 24 questionnaire packs available for analysis (24 WHOQOL-BREF questionnaires and 42 FROM-16 questionnaires).

The mean number of family members that participated within each family was 1.75 (mode = 1, median = 1) (Figure 1). The range was 1–4.

Table 1 shows participant demographics. One family member did not answer the demographic questions at the start of the FROM-16 questionnaire, so the information represents 24 patients and 41 family members. The patients’ mean age was 45 years (range: 18–71) and the family members mean age was 50 years (range: 18–94).

The median time it took patients to complete the WHOQOL-BREF questionnaire was 5–10 min (range from two minutes to one week). One patient did not provide this information and so was excluded from this analysis. Five of the 23 patients (21.7%) required assistance with completing their questionnaire.

One patient did not answer one of the WHOQOL-BREF questions in the Environment domain. We calculated the mean Environment score based on the other participants’ answers in this domain, and then used this score to complete the question.

All patients rated their QoL as either ‘Very poor’, ‘Poor’, or ‘Neither poor nor good’ in Question 1 of the WHOQOL-BREF. Similarly, they were either ‘Very dissatisfied’ ‘Dissatisfied’, or ‘Neither satisfied nor dissatisfied’ with their health, measured using Question 2. The responses to both questions had a Spearman’s rank correlation coefficient (*r_s_*) of 0.50 [*p*(2-tailed) = 0.03], showing a statistically significant, strong correlation between patient QoL and health satisfaction.

The mean scores of parents, children and partners and spouses (Table 2) are very similar (range 18.0–20.1). Of the other respondents, only five siblings and grandparents responded, with a mean score of 27.0.

Family members, on average, scored 8.8 (max = 12) in the FROM-16 emotional domain, and 11.1 (max = 20) in the personal and social life domain (Table 3). The average overall FROM-16 score was 19.9 (*n* = 42). Family members are greatly affected by the ME/CFS, with no floor effect in either of the FROM-16 domains.

Table 3 shows the mean and median score for each of the 16 FROM-16 questions, and the rank of each question, based on mean scores. The median score was 2 for 6 questions: these were (given in descending order of score magnitude): questions 1 (worried), 10 (family activities), 4 (frustrated), 3 (sad), 11 (holidays), 15 (expenses). The median score for the other 10 questions was 1.

For patients, the average score for the physical health domain (21.8) in the WHOQOL-BREF was significantly lower compared to each of the other domains (psychological = 40.9, social relationships = 40.8, environment = 54.7, *n* = 24) (Table 4).

A statistically significant correlation [*p*(2-tailed) = 0.05] was found between the QoL of the patient and that of the family members, calculated from Question 1 of the WHOQOL-BREF and the mean FROM-16 total scores (*r_s_* = −0.41) (Table 5). There was a negative correlation due to the different scoring directions of the two questionnaires.

Statistically significant, negative correlations were found between Physical Health (domain 1) of the WHOQOL-BREF and the mean FROM-16 total score and between the Environment (domain 4) and mean FROM-16 total score. Table 5 shows the Spearman’s rank correlation coefficients for the domains with the 2-tailed probability level.

## 4. Discussion

Our results confirm ME/CFS has a major negative impact on both the patient and their family members’ QoL. There was no floor effect in any of the FROM-16 responses, exemplifying some degree of impact on every single family member who participated.

With a mean total FROM-16 score of 19.9, the negative impact on family members’ QoL was significantly higher (mean = 19.9 SD = 7.2 *n* = 42) compared to previous FROM-16 scores of family members of patients with 25 other diseases (mean = 12.3, SD = 7.5, *n* = 120, *p* < 0.001) [8].

The statistically significant correlation between Question 1 of the WHOQOL-BREF and the mean FROM-16 total scores, confirms our hypothesis that in ME/CFS patients, a poorer QoL impacts greatly on their family members’ QoL. Patient’s family members’ QoL is an important concept and has shown to be adversely affected in a wide range of medical specialities, from dermatology [9] to oncology [10] whereas in certain conditions such as urinary stones, there has been a negligible impact on family members [11].

From this low number of respondents, there appears to be no obvious signal suggesting that there might be differences in QoL impact experienced by relatives of different relationship with the affected person. However, the few siblings and grandparents scored more highly. Mothers and female partners/spouses scored marginally higher than fathers and male partners/spouses. The minimal clinically important score difference (MCID) for FROM-16 has not yet been determined, but the MCID of a measure is usually in the order of 10–15% of the maximum score for that measure, i.e., a score change here of three to five. If that is the case, the differences between mean scores of parents, children and partners and spouses would not have reached the MCID level. However, the mean scores for the individual FROM-16 questions reveal which aspects of family member’s lives were most affected by having a family member with ME/CFS: these were worry, family activities, frustration and sadness. This emphasises the importance of providing appropriate psychological support and practical advice, including financial advice, to family members.

The finding that, for patients, the average score for the physical health domain was significantly lower compared to each of the other domains has also been shown within other ME/CFS research studies using the Medical Outcomes Study Short Form-36 Health Survey. All scores regarding physical function were significantly lower when compared to healthy controls and other illness groups [12,13]. A strong correlation was found between the WHOQOL-BREF ‘Physical Health’ domain and the mean FROM-16 total scores.

Also highlighted in our results, was a correlation between the WHOQOL-BREF ‘Environment’ domain and the mean FROM-16 total scores. Weak correlations with the other WHOQOL-BREF domains (Psychological and Social Relationships) and the mean FROM-16 total scores were expected as these aspects of QoL are not fully addressed in the FROM-16 questionnaire.

Limitations of this study include that questionnaires were not anonymous within each family—patients could see what their family members had answered and vice versa. Consequently, some participants may have over- or understated some of their responses. Self-reported questionnaires also carry their own risk of bias, for example, answering questions with socially-desirable answers. Some families had more than one family member with ME/CFS. This could confound the results. Different domains were measured between the questionnaires, therefore some of the domains were not expected to significantly correlate with the FROM-16 scores. The recruitment method used creates several biases that must be taken into account when interpreting the data. Respondents had to be motivated to be a member of a patient support group and were also part of an online environment. It is possible that these biases may have resulted in those patients and family members who were more highly educated, resourceful and dissatisfied as well as more severely affected were over-represented. This was a limitation noted by Hvidberg et al. who received questionnaire responses from 105 participants on a health related QoL (HRQoL) via the national ME/CFS patient association [3].

The median time of 5–10 min for patients to complete the WHOQOL-BREF questionnaire demonstrates that this could be a useful tool in a clinical or outpatient setting.

Positive aspects of the study included that this was the first attempt to measure the impact of adult ME/CFS on the QoL of family members using validated questionnaires. If indeed more severely affected patients are over-represented in this study, this would be an important advantage, as such patients are often poorly represented in studies, and the burden of ME/CFS may therefore be significantly underestimated [14].

## 5. Conclusions

To the best of our knowledge, this is the first study to measure the QoL in family members and adult patients with ME/CFS using validated questionnaires. There is a significant correlation between patient’s QoL and their family members’ QoL. This was a small exploratory study but provides sufficient evidence to support larger scale research to provide more robust evidence.

At present there is very little support available to family members of patients with ME/CFS. This study provides evidence of the major impact that this condition has on the QoL of family members. This lays down a challenge to the health care services to address these issues and to identify ways in which the secondary impact of ME/CFS may be alleviated.

## Figures and Tables

**Figure 1 medicina-57-00043-f001:**
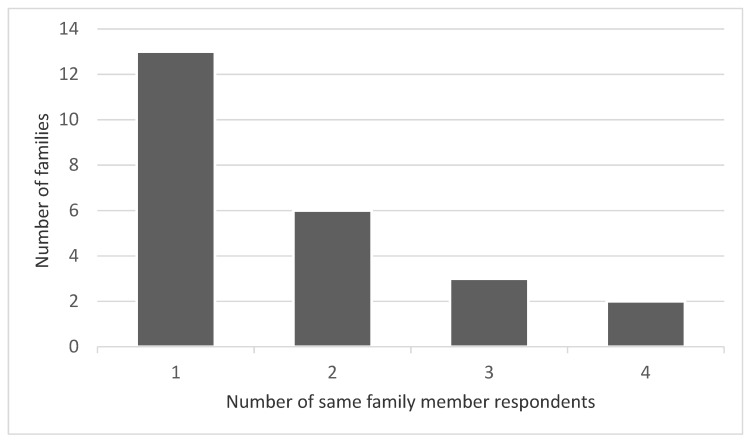
Bar Graph to illustrate the number of family members who completed the FROM-16 (Family Reported Outcome Measure-16) questionnaires within each family (*n* = 42).

**Table 1 medicina-57-00043-t001:** Patient and family member demographics.

**Patient Demographics (*n* = 24)**		
	**Frequency**	**Percentage (%)**
**Gender:**		
Female	18	75
Male	6	25
**Marital Status:**		
Single	6	25
Married	13	54
Living as married	3	13
Separated	1	4
Divorced	1	4
Widowed	0	0
None at all	0	0
Primary School	0	0
Secondary School	6	25
Tertiary	18	75
**Family Member Demographics (*n* = 41)**		
	**Frequency**	**Percentage (%)**
**Gender**		
Female	18	43.9
Male	23	56.1
**Relationship to Patient:**		
Mother	11	26.8
Father	4	9.8
Spouse	12	29.3
Partner	4	9.8
Son	3	7.3
Daughter	2	4.9
Brother	2	4.9
Sister	1	2.4
Grandmother	1	2.4
Grandfather	1	2.4

**Table 2 medicina-57-00043-t002:** Mean FROM-16 (Family Reported Outcome Measure-16) scores of family members.

	Respondents	Number of Replies	Mean FROM-16
1	Parents	15	20.1
2	Children	5	18.0
3	Partners and Spouses	16	19.1
4	Siblings and grandparents	5	27.0

**Table 3 medicina-57-00043-t003:** The rank of each question based on the mean score and the median score of each question.

Rank	1	2	3	4	5	6	7	7	9	10	10	12	13	13	15	16
Question No.	1	10	4	3	11	15	5	6	2	7	14	13	12	16	8	9
Mean FROM-16 Score	1.80	1.76	1.70	1.60	1.51	1.34	1.29	1.29	1.10	1.07	1.07	0.96	0.86	0.86	0.83	0.76
Median FROM-16 Score	2	2	2	2	2	2	1	1	1	1	1	1	1	1	1	1

**Table 4 medicina-57-00043-t004:** Average results from both the FROM-16 (Family Reported Outcome Measure-16) and WHOQOL-BREF (Abbreviated World Health Organisation Quality of Life) questionnaires.

Questionnaire	Domain	Mean	Sample Standard Deviation	Median	Range	Floor Effect (%)	Ceiling Effect (%)
**FROM-16** **(*n* = 42)**	Emotional (max 12)	8.8	2.7	9.5	2–12	- **	16.7
	Personal and social life (max 20)	11.1	5.2	11.5	1–20	-	2.4
	Overall score (max 32)	19.9	7.2	20.5	3–31	-	-
**WHOQOL-BREF** **(*n* = 24)**	Physical health *	21.8	12.5	19	0–56	4.2	-
	Psychological *	40.9	14.7	44	13–69	-	-
	Social relationships *	40.8	24.5	37.5	0–100	4.2	4.2
	Environment *	54.7	14.1	56	19–94	-	-

* Transformed score (max 100)]. The maximum FROM-16 total score is 32. The emotional and personal and social score are out of 12 and 20 respectively. ** Floor effect – all family members recorded some impact on quality of life

**Table 5 medicina-57-00043-t005:** Spearman’s rank correlation coefficient and probability level of correlation between the mean FROM-16 total scores and each of the WHOQOL-BREF domains.

WHOQOL-BREF Domain Correlated with the Mean FROM-16 Total Score.	Spearman’s Rank Correlation Coefficient (*r_s_*)	*p*(2-Tailed) Test
**Domain 1:** Physical Health	−0.510	0.01
**Domain 2:** Psychological	−0.092	0.67
**Domain 3:** Social Relationships	−0.323	0.12
**Domain 4:** Environment	−0.453	0.03

## Data Availability

The data presented in this study are available on request from the corresponding author. The data are not publicly available, patients have given consent for use of their anonymous data in this study.

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
