# Peer review of "Myalgic Encephalomyelitis/Chronic Fatigue Syndrome (ME/CFS): Major Impact on Lives of Both Patients and Family Members"

_medicina, 2021, doi:10.3390/medicina57010043_

Round 1

Reviewer 1 Report

As a ME/CFS family member and researcher into the biological basis of the illness this is a manuscript that resonates strongly with me. I have had 30 years observing the effects of the illness in 2 members of my family and of course experienced the effects and emotions as a father and spouse of affected members. Having just published 2 papers on the molecular changes in ME/CFS patients affirming the disease has a strong biological basis I have been overwhelmed by patient and family responses to my affirming their illness.

This is why this paper that attempts to quantitate the effects of the illness on patients and their family members is so important - it is a step to enhancing the societal understanding of the huge impacts of the illness on these families - in a social climate where this is trivialised and doubted by many still in the health system, and by social services that are unsupportive of severely debilitated patients in many countries. Moreover, it is such a difficult disease often for close family members to understand, and  particularly why their affected family member 'does not get well'. 

I was interested to see the respondents were mainly tertiary educated whereas perhaps the greatest need for understanding is among families with less capacity to access all the latest information on the illness.

It is interesting to see that physical health rated so dominatly compared with the other criteria, particularly social realtionships which are often so restricted. (Table 4).

The correlation with poor QoL and impact on familty members is important as it highlights the spread of the effects of this disease beyond the person affected.  The prominence of 'worry' 'frustration' and 'sadness' in family responses certainly resonated with me.

I read the limitations section with interest- 'respondents have to be motivated' - I have found most patients and families affected are highly motivated to do anything that might improve their situation, but  being part of an online environment is highly relevant - it does split the affected into two groups essentially. While I acknoweledge the questionnaires would not be anonymous in the families, most patients and family members would each have such definite opinions based on their minute by minute observations of the ongoing disease that  I think this would be less of a limitation in this case.

On my teaching now of medical students about this illness I frequently get asked 'What should I do if I meet a potential ME/CFS  patients ' My response is validate their illness even if you do not understand it, and appreciate there will be a 'traumatised' family in the shadows surrounding the person that needs help and counselling as well.

I thank the authors for documenting the effects on the illness on patients and families in this objective manner as it is an important  step to their gaining improved care from their health professionals,  and one would hope also social services.

Author Response

Thank you for reviewing our work. Thank you for taking the time to make comments and expand on some of the discussion points. We appreciate the response that some of the limitations that we have identified are not likely to be major limitations in this study. 

Thank you again for your review.

Reviewer 2 Report

At lines 154-156, the sentence “The statistically significant correlation between Question 1 of the WHOQOL-BREF and the mean FROM-16 total scores, confirms our hypothesis that ME/CFS patients with a poorer QoL impacts greatly on their family members’ QoL” is rather confused. In the subordinate clause “… that ME/CFS patients with a poorer QoL impacts greatly on their family members’ QoL”, the subject is ‘a poorer QoL’, not ‘ME/CFS patients’. This could perhaps be rephrased as follows: “… that in ME/CFS patients, a poorer QoL impacts greatly on their family members’ QoL.”

At lines 190-192, the statement is made that “It is possible that these biases may have resulted in those patients and family members who were more highly educated, resourceful and dissatisfied as well as more severely affected were over-represented”, while at lines 199-201 it is stated that “An important advantage of including more severely affected patients is that they are often poorly represented in studies”. This carries an implication that more severely affected patients were indeed over-represented in the study, whereas at lines 190-192 this is only suggested as a possibility, and no empirical evidence is presented to confirm that this is indeed the case. This could be overcome by rephrasing the sentence at lines 199-201, for example so that it would read “If indeed more severely affected patients are over-represented in this study, that would be an important advantage, as such patients are often poorly represented in studies”.

Author Response

Thank you for reviewing our work.

Lines 154-156: We have changed the wording as you suggested to improve clarity of message.

Lines 199-201: We have changed the wording to ensure the suggestion does not appear to be evidence based.

Thank you again for your review.